# Thermal, Mechanical and Tribological Properties of Gamma-Irradiated Plant-Derived Polyamide 1010

**DOI:** 10.3390/polym15143111

**Published:** 2023-07-21

**Authors:** Maiko Morino, Yosuke Nishitani, Tatsuya Kitagawa, Shinya Kikutani

**Affiliations:** 1Department of Mechanical Engineering, Graduate School of Engineering, Kogakuin University, 2665-1 Nakano, Hachioji 192-0015, Tokyo, Japan; 2Department of Mechanical Engineering, School of Engineering, Kogakuin University, 2665-1 Nakano, Hachioji 192-0015, Tokyo, Japan; 3STARLITE Co., Ltd., 2222 Kamitoyama, Ritto 520-3004, Shiga, Japan

**Keywords:** plant-derived polyamide, gamma-irradiation, cross-linking, thermal properties, mechanical properties, tribology

## Abstract

In this study, we investigated the influence of the gamma-irradiation dose and the addition of the cross-linking agent (triallyl isocyanurate (TAIC)) on the thermal, mechanical and tribological properties of plant-derived polyamide 1010 (PA1010). PA1010 and PA1010/TAIC were extruded using a twin screw extruder and injection molded. These specimens were then irradiated with gamma-ray in air with doses of 20 and 50 kGy. After gamma-irradiation, the specimens were heat-treated to remove the free radicals generated in the polymer. The combination of gamma-irradiation and the addition of TAIC significantly changed the crystal structures of PA1010. Glass transition temperature increased with the addition of TAIC and, in particular, with increasing gamma-irradiation dose. Moreover, PA1010/TAIC showed a rubbery plateau originating from cross-links by gamma-irradiation, which was observed in the temperature regions above the melting point in DMA measurements. Mechanical properties such as strength, modulus and hardness, and tribological properties such as frictional coefficient, specific wear rate and limiting *pv* (pressure *p* × velocity *v*) value of PA1010 improved with change in the internal microstructure with the gamma-irradiation and addition of TAIC.

## 1. Introduction

The field of polymeric tribomaterials for mechanical sliding parts is recently seeing a significant increase in interest in biomass polymers such as polylactic acid (PLA) and plant-derived polyamide (PA) [1,2,3,4,5,6]. This is due to the fact that biomass polymers have the potential to reduce environmental impacts related to petroleum-based plastics, including oil depletion, waste problems, and marine plastic litter. In particular, plant-derived polyamide 1010 (PA1010) is a 100% biomass-derived polymer composed of sebacic acid and decamethylenediamine obtained from castor oil [7,8,9]. As castor oil is derived from the seeds of inedible castor beans, there is no competition with human food consumption. Moreover, PA1010 is a kind of engineering plastic and semi-crystalline polymer and has unique performances such as high heat resistance, high mechanical strength, flexibility and low water absorption among biopolymers. The main application areas of PA1010 include automobiles (fuel tubing), electric/electronic devices, sporting goods (ski boots), clothing (fasteners), and everyday items (toothbrushes). Furthermore, since PA1010 is a non-biodegradable polymer, it can be applied to actual functional products that require durability. On the other hand, PA1010 has inferior mechanical (strength, modulus and hardness) and tribological properties (low friction, high wear resistance and high limiting *pv* (value) compared to petroleum-derived engineering plastics such as polyamide 6 (PA6), polyoxymethylene (POM), etc. Here, the *pv* value expresses the apparent surface pressure *p* multiplied by sliding velocity *v*. This *pv* value is often used to determine the usable operating range of the tribomaterials for the design and selection of plastic sliding bearings application. As surface pressure and sliding velocity increase under sliding conditions, there is a limit value (criteria) at which the tribomaterial can withstand the surface pressure and sliding velocity and thus breaks or melts. This limiting value is called the limiting *pv* value and shows the relationship *pv* = *c* (constant value) [9,10,11,12]. The higher this limiting *pv* value is, the higher will durability, heat resistance and wear resistance be.

In order to effectively use plant-derived PA1010 in the industry for polymeric tribomaterials, it is necessary to further enhance its mechanical and tribological properties. There are various methods of further enhancing the tribological properties of polymers, one of which is filling with fillers, solid lubricants and reinforcement fibers. Several studies have reported the results of the research on molybdenum disulfide (MoS_2_) filler and carbon fiber (CF) reinforced PA1010 composites [13], that of copper oxide (CuO) filler and CF reinforced PA1010 composites [14,15], and that of zinc oxide particles and whiskers filled PA1010 composites [16]. In our previous studies, we investigated the effect of surface treatment of fiber on the tribological properties of natural fiber (NF), namely hemp fiber, jute fiber, ramie fiber and sisal fiber, reinforced plant-derived PA1010 biomass composites [9,17,18,19,20,21,22,23,24]. We found that filling with NF and surface treatment of NF improves the tribological properties of plant-derived PA1010. However, for the application of actual functional products such as gear, bearing, cum, seal, etc., it is essential to further enhance the tribological properties of plant-derived PA1010. The enhancement methods of polymer tribology other than filler, solid lubricant and reinforcement fiber-filled polymer composites are polymer alloys/polymer blends, impregnation/application of lubricants, chemical modification, surface coating, surface modification with electron beam/gamma (γ-ray) irradiation [25].

In this study, we propose the use of gamma (γ-ray) irradiation to enhance the tribological properties of plant-derived PA1010. Gamma-irradiation has long been reported in many studies to enhance the tribological properties, in particular, wear resistance, of some plastics, such as high-density polyethylene (HDPE) [26], ultra-high molecular weight polyethylene (UHMWPE) [27,28,29,30,31,32,33], polytetrafluoroethylene (PTFE) [34,35,36], poly (ether ketone) (PEEK) [36,37,38], etc. Especially with a focus on the application of joint prostheses, gamma-irradiation has been investigated in a number of studies for its ability to enhance the wear resistance of UHMWPE. Oonishi et al. investigated the enhancement of the wear resistance of UHMWPE for joint prostheses by gamma-irradiation since the 1970s and have revealed that the wear amount of UHMWPE decreases due to cross-linking by gamma-irradiation and also the optimum dose of gamma-irradiation conditions for low wear properties [27,28,29,30,31]. Fundamentally, the mechanisms of enhancement of wear properties by gamma-irradiation may be attributed to the induction of cross-linking in the molecular chain, although gamma-irradiated polymers may also undergo chain scission and oxidative degradation.

On the other hand, PA1010 is a cross-linking polymer made by gamma-irradiation, and there have been some investigations on the gamma-irradiation effect of PA1010 on internal structures such as cross-linking crystal structures [39,40,41,42,43,44]. Zhang et al. studied the effect of gamma-irradiation on the neat PA1010 and PA1010 containing difunctional cross-linking agent, which is N, N′-bis-maleimide-4, 4′-biphenyl methane (BMI) [39,40,44]. They demonstrated that the addition of BMI in PA1010 changes the internal structure, such as crystal structure and promotes cross-linking of PA1010 molecules by gamma-irradiation, thereby reducing the radiation dose required. It has also been reported that during irradiation, the presence of BMI markedly changes the melting and crystallization characteristics of PA1010. However, although the effects of gamma-irradiation on the thermal properties and crystal structure of PA1010 have been published as mentioned above, to our knowledge, there are only a few studies that have experimentally investigated the enhancement of tribological properties of plant-derived PA1010 by gamma-irradiation. Furthermore, to our knowledge, there are no experimental data on the effects of other cross-linking agents on the various properties of PA1010 by gamma-irradiation.

In most studies, when polymers such as UHMWPE and PTFE are irradiated with gamma rays to improve their mechanical properties and wear resistance, the irradiation doses are often high doses in the range from several hundred kGy to several thousand kGy. Oonishi et al. investigated the optimum dose of gamma-radiation heavy doses, ranging from 0 to 2000 kGy, to low-wear polyethylene in total hip prostheses [29]. They concluded that the optimum dose of gamma-radiation is around 2000 kGy. Briscoe et al. studied the friction and wear of gamma-irradiated PTFE in the range from 0 to 1000 kGy [34]. They found that all the changes produced are a function of the exposure, but most of the effects are fully manifested by 200 kGy. Feng et al. also investigated the effect of gamma irradiation on the structure of PA1010 and BMI/PA1010 at high irradiation doses in the range of 0 to 12,000 kGy. However, it should be necessary to consider further energy efficiency and conservation in future studies in order to realize a sustainable society. Several studies have investigated the effect of low irradiation doses in the range of less than 100 kGy on the mechanical and tribological properties of UHMWPE. For example, Jones Jr. et al. [45], Simis et al. [33] and Sreekanth et al. [46] reported on the effects of low gamma-irradiation doses of less than 100 kGy on the mechanical and tribological properties of UHMWPE. These studies found that these properties are improved with low gamma-irradiation doses of less than 50 kGy, albeit slightly. Therefore, in this study, we also investigated whether the thermal, mechanical, and tribological properties of PA1010 can be improved in low gamma-irradiation doses, such as 20 kGy and 50 kGy, by combining TAIC, in order to realize the energy efficiency and conservation.

This study aimed to experimentally investigate the thermal, mechanical and tribological properties of gamma-irradiated plant-derived polyamide 1010. We especially studied the influence of gamma-irradiation dose and addition of cross-linking agent, which is triallyl isocyanurate (TAIC), on these properties of plant-derived PA1010.

## 2. Materials and Methods

### 2.1. Materials

Plant-derived PA1010 (Vestamid Terra DS16, Polyplastic-Evonic Corporation, Tokyo, Japan) was used as the main polymer in this study. This PA1010 is a completely biomass-derived polymer based on sebacic acid and decamethylenediamine obtained from castor oil [7,9]. Triallyl isocyanurate (TAIC; TAIC M-60, Mitsubishi Chemical Corporation, Tokyo, Japan), in which TAIC (triallyl isocyanurate) is impregnated (retained) in calcium silicate, was used as an agent to promote cross-linking. The triallyl isocyanurate content in TAIC M-60 was 60 wt.%. The weight fraction of TAIC was fixed at 1 wt.%.

### 2.2. Processing and Gamma-Irradiation

PA1010 and TAIC were dry-blended in a bottle and then dried for 12 h at 80 °C in a vacuum oven. Neat PA1010 and the mixture of PA1010 and TAIC (PA1010/TAIC) were melt-mixed at 220 °C and 85 rpm using a twin-screw extruder (TEX30HSS, Japan Steel Works, Ltd., Tokyo, Japan). After mixing, the extruded strands of PA1010 and PA1010/TAIC were cut into 5 mm pieces using a pelletizer and dried again for 12 h at 80 °C in a vacuum oven. Various shaped test pieces were injection-molded by an injection-molding machine (NEX30IV-2EG, Nissei Plastic Industrial, Co., Ltd., Nagano, Japan). These injection-molding conditions were performed as follows: 220 °C for nozzle temperature, 220 °C for cylinder temperature, 40 °C for mold (cavity) temperature, and 13 cm^3^/s for injection rate. These injection-molded specimens were irradiated with gamma rays (Cobalt-60) (Koga Isotope Ltd., Shiga, Japan) in an air atmosphere with doses of 20 and 50 kGy. Gamma-irradiation was performed using a gamma-irradiation system (Nordion Gamma Irradiators JS-8500, Nordion Inc., Ottawa, ON, Canada) with Cobalt-60 as a source of radiation. The injection-molded specimens were loaded into a special aluminum irradiation container, carried into the irradiation chamber by the conveyor, and repeatedly irradiated until the target integrated dose was reached. The integrated dose was confirmed by measuring the dose of the dosimeter element attached to the specimen and irradiated in accordance with ISO 11137 [47]. After gamma-irradiation, the specimens were heat-treated for 2 h at 100 °C to remove the free radicals generated in the polymer. In accordance with JIS K 6920-2, these test specimens were kept in desiccators for at least 24 h at 23 °C after injection-molding or gamma-irradiation. The code, composition of PA1010 and TAIC, and gamma-irradiation dose in this study are listed in Table 1. Figure 1 shows the photographs of various gamma-irradiated PA1010 specimens.

### 2.3. Experimental Method

Two types of thermal property tests were conducted in this study: differential scanning calorimetry (DSC) and dynamic mechanical analysis (DMA). DSC measurements were conducted by DSC equipment (STAR6000, SII Nanotechnology Inc., Tokyo, Japan). The samples used for the DSC were cut from pellets after mixing and gamma-irradiation into small pieces weighing 10 mg. The samples were scanned from 0 to 230 °C with a constant heating and cooling rate of 10 °C/min in a nitrogen (N_2_) atmosphere. The DMA measurement was carried out using a rheometer (ARES-G2, TA Instruments Japan, Inc., Tokyo, Japan) capable of performing linear DMA in the solid state. The DMA measuring mode was tension mode. The DMA measurement was evaluated as a function of temperature from −100 to 225 °C in an N_2_ atmosphere with a constant heating rate of 2 °C/min and tensile fixture at a frequency of 10 Hz. The strain amplitude was set at 0.05%.

Three types of static mechanical property tests were investigated in this study: tensile tests, three-point bending tests, and Durometer hardness tests. Tensile tests were carried out with dog-bone samples on a universal tester (Strograph V-10, Toyo Seiki Seisaku-Sho, Ltd., Tokyo, Japan). The tensile tests were conducted at a crosshead speed of 50 mm/min and at room temperature. Three-point bend tests were determined using coupon samples on the same universal tester V-10. The bending tests were conducted at a crosshead speed of 2 mm/min and at room temperature. The durometer hardness test was carried out using plate samples on a digital hardness tester (SD-C, ASTM type *D*, Toyo Seiki Seisaku-Sho, Ltd., Tokyo, Japan) and at room temperature.

Tribological properties were carried out using a ring-on-plate-type sliding wear tester (EFM-III-EN, Orientec, Co. Ltd., Ibaraki, Japan) at room temperature in accordance with JIS K 7218a. A carbon steel (S45C) ring was used as a metal counterpart. Two types of tribological tests were conducted in this study: constant normal load and constant sliding velocity test (normal load *P* of 140 N, sliding velocity *v* of 0.2 m/s, and sliding distance *L* of 600 m) and limiting *pv* (pressure *p* × velocity *v*) value were measured by the step load method (initial normal load *P*_0_ of 50 N, sliding velocity *v* of 0.3 m /s, and step load *P_s_* of 25 N/3 min). The experimental methods used were DSC, DMA, tensile, and three-point-bend. Durometer hardness and sliding wear tests are the same as those in our previous articles [8,9,24]; details are omitted in this article. To understand the friction and wear mechanism, the wear debris after the sliding wear test was observed using a scanning electron microscope (SEM; JSM6360LA, JEOL Ltd., Tokyo, Japan). The surfaces of all samples for SEM observation were sputter-coated with osmium (Os).

## 3. Results and Discussion

### 3.1. Differential Scanning Calorimetry Analysis

The influence of gamma-irradiation dose and addition of TAIC on crystallization behavior such as crystallization temperature *T_c_*, heat of crystallization Δ*H_c_*, melting point *T_m_*, heat of fusion Δ*H_f_* and degree of crystallinity *χ_c_* of plant-derived PA1010 using differential scanning calorimetry (DSC) analysis was investigated. It is necessary to understand these thermal properties since the thermal properties of semi-crystalline polymers, such as plant-derived PA1010, have a strong influence on mechanical and tribological properties. Figure 2 shows the DSC thermograms of various gamma-irradiated PA1010 and PA1010/TAIC: 1st cooling curves from 220 °C to 140 °C (Figure 2a) and 2nd heating curves from 140 °C to 220 °C (Figure 2b). Table 2 summarizes various DSC parameters such as crystallization temperature *T_c_*, heat of crystallization Δ*H_c_*, melting point *T_m_*_1_ and *T_m_*_2_, heat of fusion Δ*H_f_*, and degree of crystallinity *χ_c_* obtained from the 1st cooling and 2nd heating DSC curves. The degree of crystallinity *χ_c_* of various PA1010 was calculated from the following equation [48]:(1)$$\chi_{c}=\frac{\Delta H_{f}}{\left(1-\phi \right)\Delta H_{f0}}$$
where Δ*H_f_* is the heat of fusion of sample obtained from DSC measurement, *ϕ* is the weight fraction of TAIC, and Δ*H_f_*_0_ is the theoretical melt enthalpy of 100% crystalline of PA1010 (244 J/g) [49,50,51].

The DSC 1st cooling curves (Figure 2a) show that although the curves of neat PA1010, PA1010_20, and PA1010_50 have almost the same shapes, these curves shift slightly toward low temperature according to the gamma-irradiation dose. On the other hand, the gamma-irradiation dose dependence of various PA1010/TAIC shows different tendencies from that of various PA1010. Specifically, although the crystallization temperature *T_c_* of PA1010 increases when filled with the TAIC, *T_c_* of PA1010/TAIC decreases dramatically with increasing gamma-irradiation dose. In contrast, Δ*H_c_* of PA1010/TAIC has complex behavior according to the gamma-irradiation dose and has a maximum peak at 20 kGy.

Next, DSC 2nd heating curves (Figure 2b) have two melting peaks each. Although the curves of neat PA1010, PA1010_20, PA1010_50 and PA1010/TAIC have almost the same shapes, those of PA1010/TAIC_20 and PA1010/TAIC_50 are significantly different from those of the former. In particular, the two melting peaks in the PA1010/TAIC_20 and PA1010/TAIC_50 curves are close to each other, with the melting peak at the higher temperature side shifting toward the low temperature and the melting peak of the lower temperature side shifting toward the high temperature by the gamma-irradiation. Moreover, the melting peak widths of PA1010/TAIC_20 and PA1010/TAIC_50 with the gamma-irradiation are rather broad than those of other materials in this study. These multiple melting peaks have been reported in some literature on PA1010 [8,49,51,52,53,54,55] and by other polyamides [48,56]. We had already explained that the melting points at the lower temperature side *T_m_*_1_ may be due to the thin lamellae formed during the DSC 1st cooling process, and the higher ones may be attributed to the melting of the thickened crystals during the heating and annealing processes, in our previous study [8]. In this study, the change in *T_m_*_1_ of various gamma-irradiated PA1010 is slight, although *T_m_*_1_ of various gamma-irradiated PA1010 exhibits a slightly complex behavior according to the gamma-irradiation dose. On the contrary, the *T_m_*_1_ of various gamma-irradiated PA1010/TAIC increases remarkably with increasing gamma-irradiation dose. However, *T_m_*_2_s of neat PA1010, PA1010_20, PA1010_50 and PA1010/TAIC are almost the same value, whereas the *T_m_*_2_s of PA1010/TAIC_20 and PA1010/TAIC_50 shift significantly toward lower temperature by gamma-irradiation. Contrary to the *T_m_*_1_ and *T_m_*_2_, the heat of fusion Δ*H_f_* and degree of crystallinity *χ_c_* of PA1010 and PA1010/TAIC decrease with increasing gamma-irradiation dose, and those of PA1010/TAIC are slightly lower than those of PA1010. This may be due to the significant change in the crystal structure of PA1010 with the addition of TAIC combined with gamma-irradiation.

It is well known that gamma-irradiated polymers may undergo chain scission and oxidative degradation as well as the induction of cross-linking in the molecular chain [36,44,52,57,58,59]. Semi-crystalline polymers, such as low-density polyethylene, high-density polyethylene and ultra-high molecular weight polyethylene, consist of a crystalline phase and an amorphous one. Lagarde et al. explained the mechanism of the change in the internal microstructures of these semi-crystalline polymers with gamma-irradiation [57]. When semi-crystalline polymers are irradiated by gamma-irradiation, it primarily undergoes chain scission and cross-linking, which occurs as a result of the formation and recombination of free radicals in the amorphous phase. In particular, when the chain scissions in the amorphous phase are generated, they fold and crystallize the molecular chain. As a result, the crystallinity of various polyethylene increases, and the melting point *T_m_* also shifts to high temperature. In contrast, the free radicals generated in the crystalline phase stay trapped due to the reduced mobility of the molecular chains in the crystalline phase. These gamma-irradiated polymers are thermal-treated higher than the melting point. This causes the free radicals to be released from the crystalline phase. Subsequently, the mobility of the molecular chains in the crystalline phase becomes active, allowing free radicals to form new cross-links. Moreover, when the polymers heat-treated above the melting point are cooled to room temperature, molecular chain folding is inhibited due to increased cross-linking, resulting in smaller crystallinity and crystal size. On the other hand, Dong et al. also reported that gamma-irradiation reduced the crystallization in some polymers, such as PA1010, PA6, and PTFE, which are irradiated at limited ranges of doses [52]. Although the gamma-radiation cross-linking of PA1010 occurs most in the amorphous phase in the same way as other semi-crystalline polymers, gamma-irradiation damage occurs in the fold surface of the lamellae due to the destruction of the crystals. These phenomena were also reported by Li et al. [60].

Based on these findings mentioned earlier, we discuss the change in the internal microstructures of plant-derived PA1010 with gamma-irradiation using various DSC parameters obtained from DSC curves in this study. *T_c_*, *T_m_*_1_ and *T_m_*_2_ of neat PA1010 do not change significantly with gamma-irradiation, although Δ*H_c_*, Δ*H_f_* and *χ_c_* of neat PA1010 decrease slightly with the gamma-irradiation dose. Thus, the internal microstructures of neat PA1010 are slightly changed with gamma-irradiation; specifically, the crystallization behavior of neat PA1010 may have been slightly suppressed and reduced. In contrast, the *T_c_* and *T_m_* of PA1010/TAIC without the gamma-irradiation shift slightly toward higher temperature, and Δ*H_c_*, Δ*H_f_* and *χ_c_* decrease with the addition of TAIC. Hence, the influence of the addition of TAIC on the DSC parameters helps to promote the crystallization behavior of PA1010. This may be partly attributed to the calcium silicate in the TAIC acting as a crystalline nucleating agent. On the other hand, the *T_c_*, *T_m_*_2_, Δ*H_f_* and *χ_c_* of PA1010/TAIC_20 and PA1010/TAIC_50 decrease dramatically with gamma-irradiation, although Δ*H_c_* and *T_m_*_1_ of PA1010/TAIC_20 and PA1010/TAIC_50 increase with gamma-irradiation. Additionally, the peaks of *T_m_*_1_ and *T_m_*_2_ of PA1010/TAIC_20 and PA1010/TAIC_50 move closer to each other compared to the peaks of other materials in this study, and also Δ*H_f_*s of these two melting peaks are lower and broader than those of other materials. Therefore, in the case of PA1010/TAIC, gamma-irradiation may cause chain scissions and also generate free radicals, i.e., cross-linking occurs in the amorphous phase. For these reasons, gamma-irradiation inhibits the crystallization behavior of PA1010/TAIC, causing *T_c_* to shift greatly to lower temperatures. On the other hand, the crystal structures of PA1010 and PA1010/TAIC change and are damaged with gamma-irradiation. Specifically, the formed lamellae are thicker due to *T_m_*_1_, shifting to higher temperatures, and the melting of the recrystallized crystals is smaller due to *T_m_*_2_, shifting to lower temperatures. In addition, the heat of fusion and degree of crystallinity is smaller due to the changed and damaged structures of the crystals. To sum up, the internal microstructures, such as the crystalline phase and amorphous one of PA1010, are strongly influenced by gamma-irradiation and the addition of TAIC. In particular, the combination of gamma-irradiation and the addition of TAIC significantly change the internal microstructures of PA1010.

### 3.2. Dynamic Mechanical Analysis

It is essential to measure the dynamic mechanical analysis (DMA) for various polymers to understand not only their mechanical behavior, such as elasticity, viscosity and viscoelasticity, and transitions in polymer but also the molecular motion and cross-linking [8,61,62,63]. Figure 3 shows the loss tangent tan *δ* (= loss modulus *E″*/storage modulus *E*′) of various gamma-irradiated PA1010 (Figure 3a) and PA1010/TAIC (Figure 3b) as a function of temperature *T* (tan *δ* vs. *T*). Figure 3 shows the storage modulus *E*′ of various gamma-irradiated PA1010 (Figure 4a) and PA1010/TAIC (Figure 4b) as a function of temperature *T* (*E*′ vs. *T*). Figure 4b also shows a magnified view of the *E*′ vs. *T* plot from 190 to 230 °C. Table 3 summarizes various obtained DMA parameters such as the glass transition temperature *T_g_*, *E*′ at 210 °C, the average molecular weight between cross-links *M_c_* and the entanglement density or the cross-links density *ν_c_*.

The tan *δ* curves of PA1010 and PA1010/TAIC have two relaxation peaks. The peaks in the higher temperature region between 50 and 70 °C show *α*-relaxation, which represents the glass transition temperature *T_g_*. The *T_g_*s of both PA1010 and PA1010/TAIC increase with the gamma-irradiation dose. These may be attributed to not only the existence of TAIC and also the change in the internal microstructures in the amorphous phase according to the gamma-irradiation, such as the change in the molecular mobility, change in intermolecular force between segments (hydrogen bond of amide groups), and induction of cross-linking. Of course, negative factors such as chain scission and reduced molecular entanglement must also be considered. However, the result of *T_g_* increasing with increasing gamma-irradiation dose suggests that the former changes are predominantly responsible for constraining the amorphous phase. These mechanisms of how the gamma-irradiated affects the *T_g_* of PA1010 and PA1010/TAIC need to be studied further by other structural analysis methods. On the other hand, the peaks in the lower temperature region around −60 °C indicate the *β*-relaxation arising from the hydrogen bonds between the PA1010 chains [8,53]. These peaks of *β*-relaxation do not shift to higher or lower temperatures with gamma-irradiation. Thus, gamma-irradiation may not play an important role in changing the hydrogen bonds between the PA1010 chains.

The storage moduli *E*′ of PA1010 and PA1010/TAIC over the whole temperature range increase with the gamma-irradiation according to the following irradiation dose order: none (0 kGy) < 50 kGy < 20 kGy. In particular, *E*′s in the leathery region of PA1010 and PA1010/TAIC, which is a complex of solid state (crystalline phase) and rubbery state (amorphous phase), *E*′s are strongly influenced by the gamma-irradiation. Here, the leathery state is a region between *T_g_* and *T_m_*. On the other hand, when *E*′ is higher than *T_m_* of thermoplastic, it generally approaches zero due to the melting of the crystalline phase, resulting in a flow state. Because of this, when *E*′ is higher than *T_m_* of PA1010, it remarkably decreases with or without gamma-irradiation. On the contrary, PA1010/TAIC shows a different behavior with and without gamma-irradiation. When *E*′ of PA1010/TAIC is without gamma-irradiation, it approaches zero as well as that of PA1010. However, *E*′ of PA1010/TAIC_20 and PA1010/TAIC_50 with the gamma-irradiation shows a plateau behavior above the melting point *T_m_*, although it rapidly decreases with increasing temperature. The *E*′ at the plateau region of PA1010/TAIC_50 is higher than that of PA1010/TAIC_20. This plateau region is considered to be a rubbery plateau arising from the entanglement of molecular chains or the cross-links [61,62,64]. In other words, PA1010/TAIC with the gamma-irradiation has network structures caused by the entanglement of molecular chains or cross-links. In the case of cross-linkable polymer including UHMWPE, the molecular weight between entanglements or the molecular weight between cross-links *M_c_* can be determined from this *E*′ at rubbery plateau regions using the rubber theory of elasticity and, specifically, are calculated from the following equations: [61,62,64]:(2)$$M_{c}=\frac{\rho_{a}RT}{G\prime }\;$$
where *G*′ is the shear storage modulus, *ρ_a_* is the density of the amorphous PA1010 (*ρ_a_* = 1.003 g/cm^3^ [65]), *R* is the gas constant, and *T* is the absolute temperature. The shear storage modulus *G*′ can be converted from the storage modulus *E*′ obtained from DMA measurement in tensile mode using Poisson ratio *ν*:(3)$$E\prime =2\left(1+\nu \right)G\prime$$

Moreover, the entanglement density or the cross-links density *ν_c_* are estimated from the following equation [64]:(4)$$\nu_{c}=\frac{\rho_{a}}{M_{c}}\;$$

Therefore, the *ν_c_* conflicts with the *M_c_*, which means that the smaller the *M_c_*, the higher *ν_c_*. From Table 3, which is the results of the *ν_c_* and the *M_c_* of PA1010/TAIC calculated from *E*′ at rubbery plateau region at 210 °C, it may be concluded that the entanglements or the cross-links of PA1010/TAIC progress with increasing gamma-irradiation dose. However, the *M_c_* of the gamma-irradiated PA1010/TAIC is one digit higher than that of gamma-irradiated UHMWPE reported by Lee et al. [62] and Xie et al. [64]. Furthermore, *ν_c_* of the gamma-irradiated PA1010/TAIC is one digit lower than that of gamma-irradiated UHMWPE reported by the same authors earlier mentioned. Consequently, the cross-links of the gamma-irradiated PA1010/TAIC are weaker than those of the gamma-irradiated UHMWPE. In conclusion, these results, such as *T_g_* and *ν_c_* obtained from DMA measurements, are expected to have a significant influence on the mechanical and tribological properties described in the next sections.

### 3.3. Mechanical Properties

This section discusses the influence of gamma-irradiation and the addition of TAIC on the mechanical properties of plant-derived PA1010. Table 4 summarizes the results of mechanical properties such as tensile, three-point bending and Durometer hardness characteristics of various gamma-irradiated PA1010 and PA1010/TAIC in this section. The tensile strength *σ_t_* and tensile modulus *E_t_* of PA1010 and PA1010/TAIC increase with gamma-irradiation, and also those of both PA1010 and PA1010/TAIC have a maximum value at 20 kGy. Moreover, those of PA1010/TAIC are higher than those of PA1010. On the contrary, the elongation at break *ε_t_* of PA1010 and PA1010/TAIC shows the opposite tendencies from *σ_t_* and *E_t_*. On the other hand, bending strength *σ_b_*, bending modulus *E_b_* and Durometer hardness *HDD* increase with increasing gamma-irradiation doses, and those of PA1010/TAIC are higher than those of PA1010. These changes in the mechanical properties may be attributed to the changes in the internal microstructure of PA1010, such as the crystalline structure, structure of the amorphous phase, and progression of cross-linking according to the gamma irradiation and the addition of TAIC. It is well known that the crystallinity *χ_c_*, glass transition temperature *T_g_*, entanglement density and cross-link density *ν_c_* are closely related to mechanical properties. In general, the strength and modulus increase and elongation at break decreases with increasing *χ_c_*, *T_g_* and *ν_c_*. However, in the study, *χ_c_*, *T_g_* and *ν_c_* of PA1010 show different tendencies with respect to gamma-irradiation. Specifically, while *χ_c_* decreases with increasing gamma-irradiation dose and addition of TAIC, *T_g_* increases with gamma-irradiation dose and the addition of TAIC. Moreover, *ν_c_* decreases with increasing gamma-irradiation dose. Since mechanical properties such as strength, modulus, and hardness of PA1010 are fundamentally enhanced by gamma-irradiation and the addition of TAIC, this suggests that the increase in *T_g_* and the progress of cross-linking have a stronger effect for enhancing the mechanical properties than the decrease in crystallinity in this study.

### 3.4. Tribological Properties

#### 3.4.1. Sliding Wear Measurement by Constant Normal Load and Constant Sliding Velocity Test

This section discusses the influence of gamma-irradiation and the addition of TAIC on tribological properties using a ring-on-plate-type sliding wear measurement device at constant normal load *P*, constant velocity *v,* and under dry conditions of plant-derived PA1010. Figure 5 shows the results of sliding wear measurement at *P* of 140 N and *v* of 0.2 m/s of various gamma-irradiated PA1010 and PA1010/TAIC: frictional coefficient *μ* (Figure 5a) and specific wear rate *V_s_* (Figure 5b). The frictional coefficient *μ* and specific wear rate *V_s_* of PA1010 decrease with the addition of TAIC. Both *μ* and *V_s_* of PA1010 and PA1010/TAIC decrease fundamentally with increasing gamma-irradiation dose, although *V_s_* of PA1010/TAIC_20 is slightly lower than that of PA1010/TAIC_50. These tendencies are similar to the improvement tendencies of mechanical properties, as earlier mentioned.

It is well known that the wear resistance of cross-linkable polymers such as UHMWPE can be improved by gamma-irradiation [27,28,29,30,31,32,33]. This is because gamma-irradiation changes internal microstructures such as crystalline structure, crystallinity, amorphous structure and degree of cross-linking. As a result, wear resistance and mechanical properties such as strength, modulus, hardness, creep resistance and fatigue resistance of UHMWPE improve with gamma-irradiation. In short, with the appropriate gamma-irradiation dose, optimized internal microstructure such as crystalline structure, crystallinity, amorphous structure and the degree of cross-linking may enhance wear resistance and mechanical properties [32]. Similarly, gamma-irradiation may change the internal structure of PA1010 and PA1010/TAIC, thereby enhancing tribological properties as well as mechanical properties.

To clarify the friction and wear mechanisms of PA1010 and PA1010/TAIC with/without the gamma-irradiation, the observation of the shape and size of wear debris generated during sliding is essential [9,24]. Figure 6 shows the SEM photographs of wear debris of various gamma-irradiated PA1010 and PA1010/TAIC collected from outside the sliding surface after the sliding wear measurement: PA1010 (Figure 6a), PA1010_20 (Figure 6b), PA1010/TAIC (Figure 6c) and PA1010/TAIC_20 (Figure 6d). The shape and size of the wear debris of PA1010 and PA1010/TAIC changed significantly with gamma-irradiation, although those of PA1010 changed slightly when filled with TAIC. Those of neat PA1010 (Figure 6a) consisted of a mixture of some long filamentary (roll) particles, which were often observed in various neat PA [9], and small granular ones; those of PA1010_20 (Figure 6b) consisted of a few short filamentary ones and many fine granular ones; those of PA1010/TAIC (Figure 6c) were similar to those of PA1010; and those of PA1010/TAIC_20 (Figure 6d) were similar to those of PA1010_20. However, the size of granular particles in the wear debris of PA1010_TAIC and PA1010/TAIC_20 was slightly bigger than those of PA1010 and PA1010_20, respectively, suggesting that the influence of gamma-irradiation on the shape and size of wear debris of PA1010 is much stronger than that of the addition of TAIC. These differences in the shape and size of the wear debris of PA1010 when gamma-irradiation and added with TAIC may be due to the change in the mode of the friction and wear mechanisms with the enhancement in mechanical properties of PA1010 by gamma-irradiation, which changes the internal microstructure of PA1010.

#### 3.4.2. Limiting *pv* Values Measured by Step Load Method

This section discusses the results of measuring limiting *pv* (pressure *p* x velocity *v*) by the step load method of gamma-irradiated PA1010 and PA1010/TAIC. These limiting *pv* values are used to determine the critical operating conditions under which the material fails and breaks, as well as to characterize the durability, heat-resistance and wear resistance of polymeric tribomaterials [9,10,11,12]. Figure 7 shows the results of measuring sliding wear by step load methods, such as the frictional coefficient *μ* and step load *P* as a function of sliding distance *L*, of various gamma-irradiated PA1010 (Figure 7a) and PA1010/TAIC (Figure 7b), respectively. *μ* of PA1010 abruptly increases with increasing *P* up to about 75 N (*L* of 150 m) and then gradually decreases with *P*. The limiting load *P_lim_*, which is just before the test pieces fractured or melted, is the following order: PA1010 < PA1010_50 < PA1010_20. On the other hand, *μ* of PA1010/TAIC also abruptly increases with *P* up to about 75 N and then decreases with increasing *P*. However, the *P_lim_* of PA1010/TAIC is the following order: PA1010/TAIC < PA1010/TAIC_20 < PA1010/TAIC_50.

This *P_lim_* is divided by the apparent contact area *A_a_* = 2 cm^2^, and the result obtained is taken as the apparent limiting contact pressure *p_lim_*. This apparent limiting contact pressure *p_lim_* was multiplied by the test velocity *v* to determine the limiting *pv* value. Figure 8 shows the relationship between the limiting *pv* value measured by the step load method and the gamma-irradiation dose of various gamma-irradiated PA1010 and PA1010/TAIC. The limiting *pv* values of PA1010 increase with gamma-irradiation and the addition of TAIC. Although the limiting *pv* values of PA1010 show a maximum peak at 20 kGy, those of PA1010/TAIC increase with the gamma-irradiation dose. These limiting *pv* values are closely related to load-bearing capacity [9,12]. The influence of the gamma-irradiation and the addition of TAIC on the limiting *pv* values of PA1010 and PA1010/TAIC are in accord with the mechanical properties of PA1010 and PA1010/TAIC listed in Table 4. Therefore, these limiting *pv* values are considered to be due to the change in the mode of friction and wear mechanisms according to the enhancement of mechanical properties of PA1010 and PA1010/TAIC as a result of the change in the internal microstructure of PA1010 and PA1010/TAIC with/without gamma-irradiation and the addition of TAIC.

## 4. Conclusions

In the present study, we investigated the thermal, mechanical and tribological properties of gamma-irradiated plant-derived polyamide 1010, especially the influence of the gamma-irradiation dose and addition of cross-linking agent (triallyl isocyanurate (TAIC)), on these properties of plant-derived PA1010. We concluded that both the gamma-irradiation and addition of TAIC strongly influenced the thermal, mechanical and tribological properties of PA1010. In particular, the combination of gamma-irradiation and the addition of TAIC was found to have the following effects; (1) Internal microstructures such as the crystalline and amorphous phases of PA1010 were strongly affected by gamma-irradiation and the addition of TAIC. Particularly, gamma-irradiation and the addition of TAIC significantly changed the crystal structures and degree of crystallinity of PA1010. (2) The glass transition temperature increased with the addition of TAIC, in particular, with increasing gamma-irradiation dose. (3) PA1010/TAIC showed a rubbery plateau originating from cross-links by gamma-irradiation, which was observed in temperatures region above the melting point in DMA measurements. (4) The combination of gamma-irradiation and the addition of TAIC significantly changed the internal microstructures of PA1010, thereby enhancing mechanical properties such as strength, modulus and hardness of PA1010. (5) These findings suggest that the increase in *T_g_* and the progress of cross-linking have a stronger effect on enhancing the mechanical properties than the decrease in crystallinity. (6) Sliding wear measurement at constant normal load and constant velocity method results showed that the tribological properties of PA1010 improved with gamma-irradiation and the addition of TAIC. Notably, gamma-irradiation and TAIC were also found to improve the wear resistance of PA1010 more than the frictional coefficient. (7) Limiting *pv* value of PA1010 by the step load method enhanced with the gamma-irradiation and addition of TAIC.

This study clearly demonstrated that the combination of gamma-ray irradiation and the addition of TAIC may serve as a promising method for improving the thermal, mechanical, and tribological properties of PA1010 compared to conventional methods, and these findings are expected to expand the application fields of biomass polymers in the future.

However, there are very few studies on the comparisons of the thermal, mechanical and tribological properties of gamma-irradiated PA1010 and those of PA1010 modified by conventional or alternative methods, such as the application of lubricants and/or surface coating. In this study, since we focused on the influence of gamma-irradiation dose and addition of cross-linking agent (triallyl isocyanurate (TAIC)) on these properties of plant-derived PA1010, it is necessary to conduct these comparative studies in the future.

## Figures and Tables

**Figure 1 polymers-15-03111-f001:**
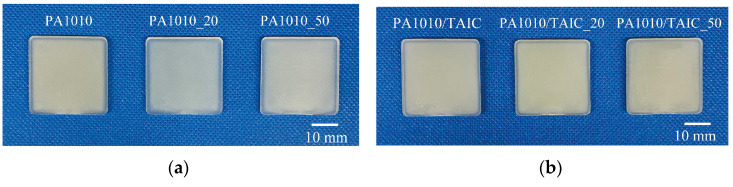
Photographs of various gamma-irradiated PA1010 specimens: (**a**) PA1010 and (**b**) PA1010/TAIC.

**Figure 2 polymers-15-03111-f002:**
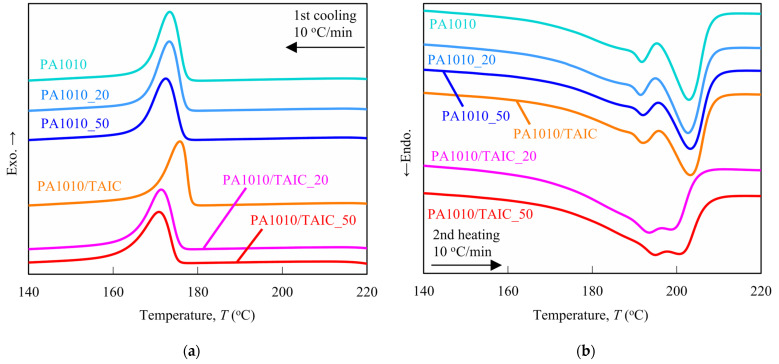
DSC thermograms of various gamma-irradiated PA1010 and PA1010/TAIC: (**a**) 1st cooling curves from 220 °C to 140 °C and (**b**) 2nd heating curves from 140 °C to 220 °C.

**Figure 3 polymers-15-03111-f003:**
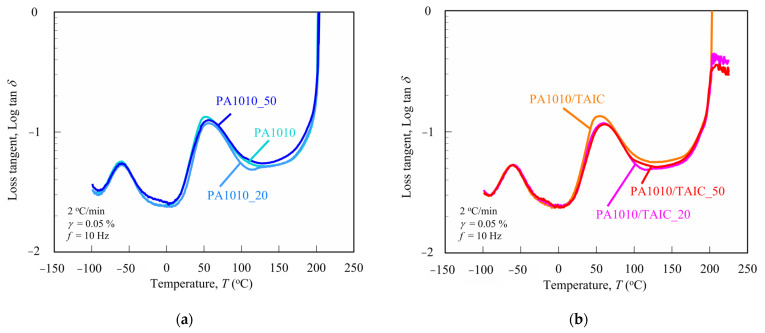
Loss tangent tan *δ* of various gamma-irradiated PA1010 and PA1010/TAIC as a function of temperature *T*: (**a**) PA1010 and (**b**) PA1010/TAIC.

**Figure 4 polymers-15-03111-f004:**
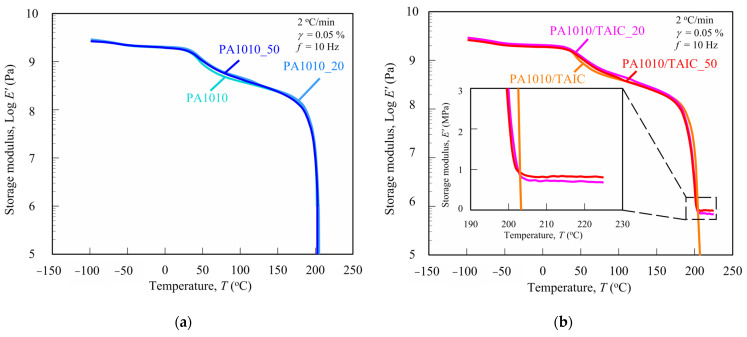
Storage modulus *E*′ of various gamma-irradiated PA1010 and PA1010/TAIC as a function of temperature *T*: (**a**) PA1010 and (**b**) PA1010/TAIC.

**Figure 5 polymers-15-03111-f005:**
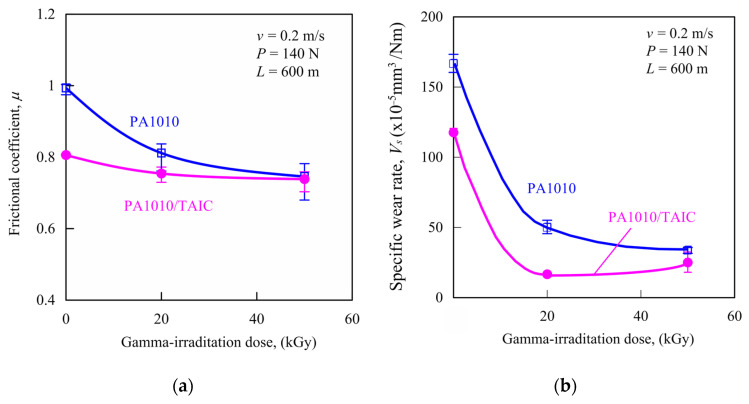
Sliding wear test by *P* of 140 N and *v* of 0.2 m/s of various gamma-irradiated PA1010 and PA1010/TAIC: (**a**) frictional coefficient *μ* and (**b**) specific wear rate *V_s_*.

**Figure 6 polymers-15-03111-f006:**
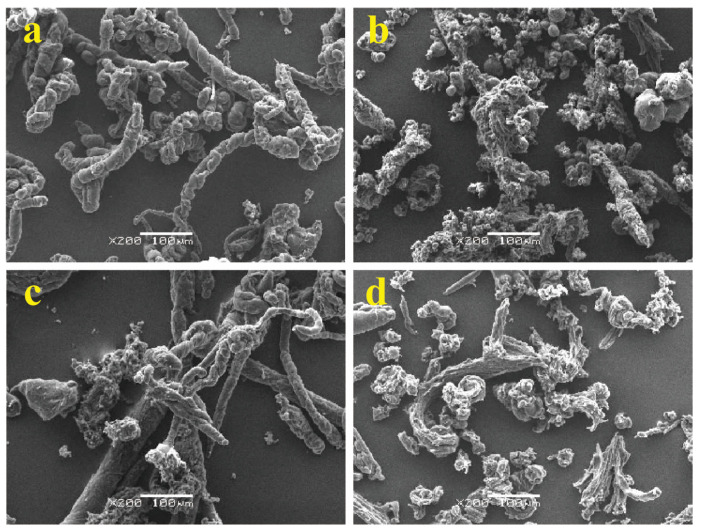
SEM photographs of wear debris of various gamma-irradiated PA1010 and PA1010/TAIC collected from the outside of sliding surface after sliding wear test: (**a**) PA1010, (**b**) PA1010_20, (**c**) PA1010/TAIC and (**d**) PA1010/TAIC_20.

**Figure 7 polymers-15-03111-f007:**
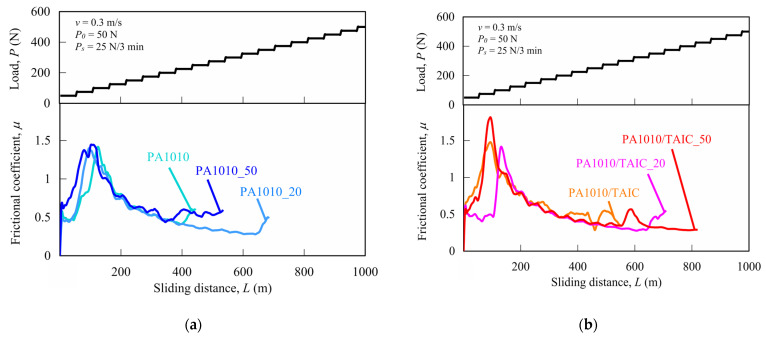
Limiting *pv* values calculated by step load method of gamma-irradiated various PA1010 and PA1010/TAIC: (**a**) PA1010 and (**b**) PA1010/TAIC.

**Figure 8 polymers-15-03111-f008:**
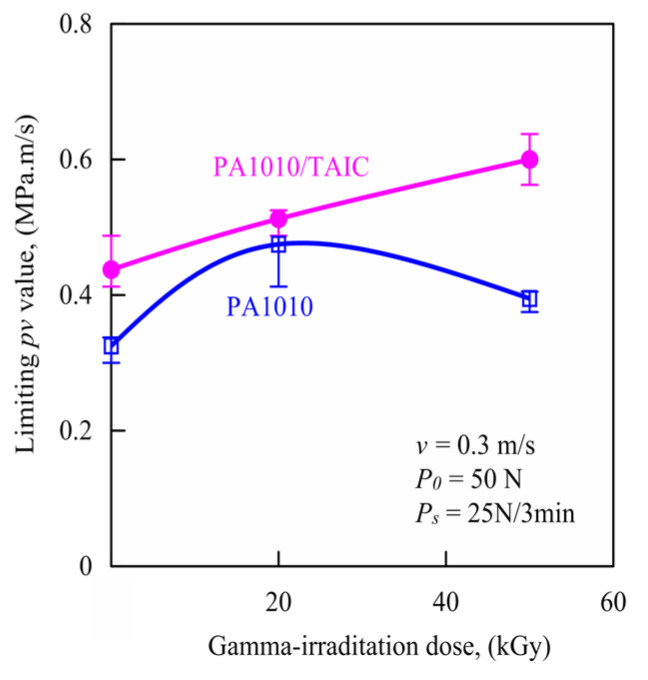
Relationship between limiting *pv* values calculated by step load method and gamma-irradiation dose of gamma-irradiated PA1010 and PA1010/TAIC.

**Table 1 polymers-15-03111-t001:** Code, composition of PA1010 and TAIC, and gamma-irradiation dose used in this study.

Code	Composition (wt.%)		Gamma-Irradiation Dose
	PA1010	TAIC	(kGy)
PA1010	100	-	-
PA1010_20	100	-	20
PA1010_50	100	-	50
PA1010/TAIC	99	1	-
PA1010/TAIC_20	99	1	20
PA1010/TAIC_50	99	1	50

**Table 2 polymers-15-03111-t002:** DSC parameters of various gamma-irradiated PA1010 and PA1010/TAIC.

	1st Cooling		2nd Heating			
	*T_c_*	Δ*H_c_*	*T_m_* _1_	*T_m_* _2_	Δ*H_f_*	*χ_c_*
	°C	J/g	°C	°C	J/g	%
PA1010	173.35	69.3	191.71	202.94	102.4	42.0
PA1010_20	173.27	69.2	191.48	202.70	98.8	40.5
PA1010_50	172.49	66.7	192.05	203.26	90.1	36.9
PA1010/TAIC	175.78	61.3	192.02	203.29	97.6	40.4
PA1010/TAIC_20	171.36	72.0	193.52	198.60	95.1	39.4
PA1010/TAIC_50	170.84	65.5	194.94	200.56	91.3	37.8

**Table 3 polymers-15-03111-t003:** DMA parameters of various gamma-irradiated PA1010 and PA1010/TAIC.

	*T_g_* °C	*E*′ at 210 °C MPa	*M_c_* at 210 °C ×10^3^ g/mol	*ν_c_* at 210 °C mol/m^3^
PA1010	50.8	-	-	-
PA1010_20	52.8	-	-	-
PA1010_50	54.8	-	-	-
PA1010/TAIC	57.8	-	-	-
PA1010/TAIC_20	60.8	0.74	16.3	61
PA1010/TAIC_50	61.8	0.81	15.0	67

**Table 4 polymers-15-03111-t004:** Mechanical properties of various gamma-irradiated PA1010 and PA1010/TAIC.

	Tensile Strength *σ_t_* MPa	Tensile Modulus *E_t_* GPa	Elongation at Break *ε_t_* %	Bending Strength *σ_b_* MPa	Bending Modulus *E_b_* GPa	Durometer Hardness *HDD*
PA1010	40 ± 1.2	1.4 ± 0.06	121 ± 5	53 ± 0.8	1.2 ± 0.04	75 ± 1.4
PA1010_20	48 ± 4.2	1.6 ± 0.19	110 ± 24	67 ± 0.5	1.2 ± 0.11	77 ± 2.4
PA1010_50	45 ± 2.9	0.9 ± 0.24	130 ± 14	62 ± 0.5	1.3 ± 0.00	78 ± 1.2
PA1010/TAIC	43 ± 2.9	1.4 ± 0.12	111 ± 15	54 ± 0.3	1.2 ± 0.07	76 ± 1.1
PA1010/TAIC_20	50 ± 0.9	1.8 ± 0.06	70 ± 15	71 ± 0.2	1.5 ± 0.00	79 ± 1.5
PA1010/TAIC_50	44 ± 1.5	1.0 ± 0.20	90 ± 18	68 ± 0.2	1.5 ± 0.11	79 ± 0.7

## Data Availability

The data presented in this study are available on request from the corresponding author.

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
