# Peer review of "Thermal, Mechanical and Tribological Properties of Gamma-Irradiated Plant-Derived Polyamide 1010"

_polymers, 2023, doi:10.3390/polym15143111_

Round 1

Reviewer 1 Report

Within the scope of this work, the authors investigated the plant-derived polyamide 1010 in terms of their thermal, mechanical, and tribological properties using before-and-after irradiation aspects. The effect of TAIC dopant in addition to varying irradiation doses were figured out in detail. In my opinion, using plant-derived polyamide possess great potential in replacement of petroleum-based ones, in order to sustain environmentally friendly concept. Yet more, the conception of performing before-and-after irradiation treatments is highly welcome for existing literature. For the mentioned reasons, this work can be evaluated as a potential publication in this esteemed journal; nevertheless, the authors should first revise the following suggestions.

1-    The authors should select fundamental keywords. I think it is too much presenting 10 keywords.

2-    In the introduction part, please ensure some application areas that utilizes plant-derived polyamide, especially in the first paragraph.

3-    In the introduction part, namely in the second paragraph, please add examples after this sentence “However, for application of actual functional products,…”.

4-    To my way of thinking, the authors should reveal their synthesized samples in a scaled photograph. With this, the readers can easily comprehend.

5-    The authors should give the details of gamma-rays irradiation measurement. The configuration including source, sample, detector, and the others should be revealed. An example can be seen below:

·         https://doi.org/10.1016/j.ceramint.2021.06.044

6-    What is the main reason for selecting 20 and 50 kGy doses? Please explain it with the available literature studies.

7-    It is difficult to check all figures in the manuscript. I could not understand whether any resolution problem exists, or any inconvenience while uploading. Please have a look at the figures one more time.

8-    The results were thoroughly presented, and the corresponding discussions were made in a satisfactory manner. Nevertheless, one may add benefits of using gamma-rays irradiation treatment on surface modification after comparing it with the conventional or alternative methods, such as application of lubricants and/or surface coating. In my opinion, this will highly strengthen the motivation of this work.

9-    The conclusion part should have a striking remark at the end of the paragraph. For example, the authors may emphasize that this study clearly indicated that gamma-rays irradiation may serve as a promising treatment for surface properties, rather than conventionally used ones. I recommend authors to consider this perspective.

Reviewer 2 Report

See attachment. Well done paper with minor revision requirements. I never heard about the pv-Value for tribological questions, so I would be glad if you can explain this.
